# A Routine Sanger Sequencing Target Specific Mutation Assay for SARS-CoV-2 Variants of Concern and Interest

**DOI:** 10.3390/v13122386

**Published:** 2021-11-29

**Authors:** Sin Hang Lee

**Affiliations:** Affiliation Milford Molecular Diagnostics Laboratory, 2044 Bridgeport Avenue, Milford, CT 06460, USA; shlee01@snet.net

**Keywords:** Delta variant, variants of concern, variants of interest, SARS-CoV-2, spike protein, nested RT-PCR, Sanger sequencing, amino acid mutations, ACE2 RBD, N-terminal domain (NTD)

## Abstract

As SARS-CoV-2 continues to spread among human populations, genetic changes occur and accumulate in the circulating virus. Some of these genetic changes have caused amino acid mutations, including deletions, which may have a potential impact on critical SARS-CoV-2 countermeasures, including vaccines, therapeutics, and diagnostics. Considerable efforts have been made to categorize the amino acid mutations of the angiotensin-converting enzyme 2 (ACE2) receptor binding domain (RBD) of the spike (S) protein, along with certain mutations in other regions within the S protein as specific variants, in an attempt to study the relationship between these mutations and the biological behavior of the virus. However, the currently used whole genome sequencing surveillance technologies can test only a small fraction of the positive specimens with high viral loads and often generate uncertainties in nucleic acid sequencing that needs additional verification for precision determination of mutations. This article introduces a generic protocol to routinely sequence a 437-bp nested RT-PCR cDNA amplicon of the ACE2 RBD and a 490-bp nested RT-PCR cDNA amplicon of the N-terminal domain (NTD) of the S gene for detection of the amino acid mutations needed for accurate determination of all variants of concern and variants of interest according to the definitions published by the U.S. Centers for Disease Control and Prevention. This protocol was able to amplify both nucleic acid targets into cDNA amplicons to be used as templates for Sanger sequencing on all 16 clinical specimens that were positive for SARS-CoV-2.

## 1. Introduction

The COVID-19 crisis has continued its pace. According to real time world statistics, as of 7 September 2021, there were >222 million cumulative human cases with >4.5 million deaths due to COVID-19 since the outbreak [1]. In the meantime, numerous amino acid mutations of the spike (S) protein of the SARS-CoV-2, the causative agent of COVID-19, are being recognized as whole genome sequencing data generated by the next generation sequencing (NGS) technology and have been used more widely for genomic surveillance [2]. Great efforts have been made to categorize these S protein amino acid mutations or substitutions into specific groups, according to their combination profiles. A few of these groups are referred to as variants in an attempt to correlate these amino acid mutation profiles with a possible increased transmissibility, increased virulence, or reduced effectiveness of vaccines against them [3,4].

The U.S. Centers for Disease Control and Prevention (CDC) has selected four variants of concern, namely the Alpha, Beta, Gamma, and Delta variants, to be closely monitored for their potential impact on critical SARS-CoV-2 countermeasures, including vaccines, therapeutics, and diagnostics. In addition, four variants of interest, namely the Eta, Iota, Kappa, and Pango Lineage B.1.617.3 variants, are being monitored and characterized [3].

As widely reported in mass media, the SARS-CoV-2 Delta variant has spread around the world [5] and is becoming the variant of most concern [6]. However, the science and procedures of how to accurately test for the Delta variant and to differentiate it from other variants for meaningful data analyses remain unclear. The CDC’s definition for the Delta variant depends on demonstration of a specific profile of amino acid mutations listed as T19R, (V70F*), T95I, G142D, E156del, F157del, R158G, (A222V*), (W258L*), L452R, T478K, D614G, P681R, and D950N, with possible additional K417N. When K417N is also detected, the variant is designated as Delta Plus [3]. The amino acid mutations with a symbol * in parenthesis (*) indicate that the mutation of this particular amino acid may or may not occur and will not affect a Delta variant designation. The amino acid mutations not in a (*) are invariable or constant mutations, which must be present in a Delta variant. However, it is not known if all laboratories performing variant testing are generating unambiguous sequencing data to verify all these 15 potential amino acid mutations before making a diagnosis of the Delta variant of SARS-CoV-2.

In March 2021, the European Centre for Disease Prevention and Control (ECDC) and the World Health Organization (WHO) jointly advised that whole genome, or at least complete or partial S-gene sequencing, should be performed to confirm infection with a specific variant for detection and identification of circulating SARS-CoV-2 VOCs [7].

However, the currently used whole genome/NGS surveillance technology for SARS-CoV-2 genomic sequencing is not always successful, particularly when there is not enough viral load in the specimen [8,9]. For example, when a series of N gene RT-qPCR-positive SARS-CoV-2 clinical laboratory specimens without an accompanying Ct value in the positive reports was processed for whole genome sequencing and viral lineage designation, a SARS-CoV-2 genomic sequence could not be obtained in 1866 of the 2045 RT-qPCR positive specimens, indicating a sequencing failure rate of 91%. In comparison, for specimens with a Ct ≤ 27, the sequencing failure rate was only 5.3%, and for those with a Ct > 27, the sequencing failure rate was 75.5% [10]. In addition, the NGS technology is also known to be associated with computational errors and biases in base-calling [11]. Target specific mutation assays are needed to identify or verify variants of concern [9] and for their differentiation from other variants.

In addition, based on information available in the public domain, even the profiles used to identify the Delta variant vary from laboratory to laboratory. According to a WHO document, the characteristic S protein amino acid mutations for the Delta variant, also known as the B.1.617.2 lineage, are T19R, G142D, 157del, 158del, L452R, T478K, D614G, P681R, and D950N [12]. Although both the WHO and CDC emphasize the presence of the T478K mutation in the Delta variant, a search of the GenBank database failed to find a SARS-CoV-2 genomic sequence containing a combination of G142D, 157del, 158del, L452R, and T478K with an intact E156 in the S gene. On the other hand, Public Health England has advised that P681R must be present in a Delta variant, but also stated that genotyping assay for B.1.617 cannot distinguish between Kappa, Delta, and the B.1.617.3 lineage, and all results with P681R are treated as probable Delta given the current dominance of this lineage [13]. Therefore, there is a need to develop a practical protocol for variant classification on all SARS-CoV-2 isolates. The ECDC and WHO suggested that the region to be sequenced should cover at least the entire N-terminal and receptor binding domain (RBD) (amino acid 1-541, 1623 bp) to reliably differentiate between the circulating variants [7].

This article introduces a simplified target amplicon sequencing assay on all nasopharyngeal swab samples, which are positive for SARS-CoV-2, to accurately determine the signature S protein amino acid mutations in the N-terminal and ACE2 receptor binding domains that are used to characterize variants of concern and of interest, according to the CDC’s definitions. In the United States, when the tests are certified under the Clinical Laboratory Improvement Amendments of 1988 (CLIA) and performed in a CLIA-certified high-complexity clinical laboratory, such as the author’s, the tests can be used as a routine diagnostic assay to assist patient management [14].

## 2. Materials and Methods

Since both the WHO and the CDC definitions of SARS-CoV-2 variants of concern and interest depend on determination of the specific profiles of amino acid mutations from K417 to N501 in the ACE2 RBD, supplemented by mutations in other regions of the 1273 amino acid chain of the spike protein, especially by those in the NTD [3,12], a brief review of these common mutations is needed in order to select the target segments of the S gene for Sanger sequencing.

GISAID automatically updates its site of hCoV-19 spike glycoprotein mutation surveillance dashboard. The updates include spike protein changes in amino acid sequences of the ACE2 receptor binding domain (RBD) newly submitted to GISAID, displayed in structures organized by the most common clades. The 24/25 August 2021 dashboard data showed the new clades (Figure 1), all of which contain mutations commonly used for Delta variant categorization.

While the GISAID hCoV-19 S protein mutation surveillance focuses on the ACE2 RBD mutations, some researchers have pointed out that the Delta variant has several unique mutations in the ACE2 RBD and the N-terminal domain (NTD) of the spike protein. The mutations in the NTD, such as T19R, G142D, E156G, F157del, and R158del, are involved in the enhanced infectivity by the BNT162b2-immune sera. The neutralizing activity of sera from vaccinated individuals, as well as convalescent COVID-19 patients, decreases for the Delta variant compared to the wild-type SARS-CoV-2 [15,16,17]. Both ACE2 RBD and NTD mutations should be evaluated [7] on all positive samples to understand the pathogenicity of the SARS-CoV-2 variants. The CDC’s classifications and definitions of SARS-CoV-2 variants of concern (VOCs) and variants of interest (VOIS) are summarized in Table 1.

### 2.1. Using Amino Acid Mutations in ACE2 RBD and NTD for Variant Determination

Based on information retrieved from the GenBank database, a sequence of 116 amino acids from T393 to Y508, highlighted yellow in Figure 2, contains the entire ACE2 RBD from K417 to N501. A sequence of 160 amino acids from M1 to Y160, highlighted green in Figure 2, covers the entire NTD whose mutations are used as additional characteristics for variant categorization [3]. Assuming the classification algorithms defined by the CDC (Table 1) to be valid and stringent, accurate determination of the mutations of the amino acids from S45 to R158 and from positions K417 to N501 should be adequate for variant categorization.

### 2.2. Patient Samples Used for Method Development

The materials used for method development were the residues of 16 SARS-CoV-2 positive nasopharyngeal swab specimens from patients with clinical respiratory infections. These were previously tested patient specimens without patient identifications and were purchased from Boca Biolistics Reference Laboratory, Pompano Beach, FL, a commercial reference material laboratory endorsed by the U.S. Food and Drug Administration (FDA) as a supplier of clinical samples positive for SARS-CoV-2 by RT-qPCR assays. According to the commercial supplier, the swabs were immersed in VTM after collection and stored in freezer at −80 °C temperature following the initial testing.

In the author’s laboratory, these 16 swab rinse specimens were proven to contain SARS-CoV-2 genomic RNA by successful bi-directional Sanger sequencing of a 398-bp N gene cDNA PCR amplicon. These 16 sequencing-confirmed positive samples were among the 30 specimens that were purchased and were initially classified as positive by RT-qPCR tests granted emergency use authorization (EUA) by the FDA for the presumptive qualitative detection of nucleic acid from the 2019-nCoV [18]. The general characteristics of these 30 swab specimens were previously published in detail elsewhere [19]. According to the commercial supplier, all these positive samples were re-tested by an EUA N gene RT-qPCR assay with Ct values ranging from 14.55 to 36.71. Nevertheless, only 16 of the 30 samples were shown to contain SARS-CoV-2 genomic RNA by partial N gene sequencing [19]. The test results of these 30 RT-qPCR positive clinical specimens collected from patients suspected of SARS-CoV-2 infection were used to fulfill the requirement for Clinical Laboratory Improvement Amendments (CLIA) certification to perform routine partial N gene Sanger sequencing for SARS-CoV-2 detection and reflex target S gene Sanger sequencing to determine variants of concern and interest. According to the FDA guidance, false results generated by RT-qPCR tests can be investigated using Sanger sequencing [20]. There are no FDA-authorized diagnostic test kits for SARS-CoV-2 variant determination.

### 2.3. RNA Extraction from Nasopharyngeal Swab Specimens

Instead of cell-free fluid samples, which are used for most RT-qPCR assays, cellular components are routinely included in the material being tested in this assay [21]. The initially published protocol was slightly modified. Briefly, about 1 mL of the residue of the nasopharyngeal swab rinse in VTM was transferred to a graduated 1.5 mL microcentrifuge tube and centrifuged at ~16,000× *g* for 5 min to pellet all cells and cellular debris. The supernatant was discarded except the last 0.2 mL, which was left in the test tube with the pellet. To each test tube containing the pellet with 0.2 mL supernatant, 200 µL of digestion buffer containing 1% sodium dodecyl sulfate, 20 mM Tris-HCl (pH 7.6), 0.2M NaCl, and 700 μg/mL proteinase K, was added. The mixture was digested for 1 hr in a heated shaker set at 47 °C. After digestion, an equal volume (400 µL) of acidified 125:24:1 phenol/chloroform/isoamyl alcohol mixture (Thermo Fisher Scientific Inc., Waltham, MA, USA) was added to each tube. After vortexing twice for extraction and centrifugation at ~16,000× *g* for 5 min to separate the phases, the liquid in the phenol/chloroform phase was pipetted out and discarded. To the remaining aqueous phase solution, 300 µL of acidified 125:24:1 phenol/chloroform/isoamyl alcohol mixture was added for a second extraction. After a second centrifugation at ~16,000× *g* for 5 min to separate the phases, 200 µL of the aqueous supernatant without any material at the interface was transferred to a new 1.5 mL microcentrifuge tube for nucleic acid purification [21].

### 2.4. PCR Primers

As reported by the CDC, nested PCR is the necessary step to generate SARS-CoV-2 cDNA amplicons to be used as the templates for Sanger sequencing [22]. The sequences, the sizes of the amplicons, and the reference location of the major primers used in this study are listed in Table 2.

### 2.5. PCR Conditions

The primary and nested RT-PCR conditions were described in detail previously [21]. This nested RT-PCR protocol has been shown to be able to amplify a single copy of target SARS-CoV-2 RNA to be used as template for Sanger sequencing [21].

### 2.6. DNA Sequencing

Sanger sequencing of the nested PCR amplicons was performed as previously described [21]. The workflow from nucleic acid extraction to variant determination by Sanger sequencing is summarized in Figure 3.

As reported previously [21], the RNase P gene selected by the CDC as the extraction control for its RT-qPCR test panel was not always amplifiable by conventional PCR for DNA sequencing. A segment of human *BRCA1* gene was chosen as the internal cellular extraction control in the current protocol. *BRCA1* gene is always present and is only found in mammalian cells [23]. Other house-keeping genes may be used instead after validation.

## 3. Results

### 3.1. The Samples Positive for N Gene Also Contained an Intact S Gene

Since mutations are widely scattered in the S protein amino acid chain among the variants (Table 1), and PCR amplification of different specific segments of the 3822-base S gene may be needed for Sanger sequencing and for the differentiation of emerging variants, it is important to confirm that all RNA extracts from the clinical specimens, which were positive for an N gene segment [19,21], also contained an intact S gene. One way to achieve this goal without performing an entire S gene sequencing was to use the SB7/SB8 nested PCR primer set to amplify a 490 bp cDNA at position 21628–22117 (Table 2) and the VF3/VF4 nested PCR primer set to amplify a 315 bp cDNA at position 24913–25227 (Table 2) on all 16 nasopharyngeal samples that previously tested positive for the N gene. The representative parts of these two sequences from one sample are shown in Figure 4.

The upper panel of Figure 4 was excised from an electropherogram of a 490 bp amplicon sequence of the S gene defined by the nested PCR primers SB7 and SB8 (Table 2). The computer-generated sequence has been converted to a 5′-3′ reading that was re-typed under the upper electropherogram with the last three bases of the SB7 forward PCR primer “CAC” underlined. The number 21646–21717 indicates the position of this segment of sequence in the SARS-CoV-2 genome. The letter “T” in red means that the wild-type nucleotide in this position has undergone a nonsynonymous mutation causing an H49Y amino acid mutation (CAT > TAT). The lower panel was excised from an electropherogram of a 315 bp amplicon sequence of the S gene defined by the nested PCR primers VF3 and VF4 (Table 2). The computer-generated sequence in a 5′-3′ reading direction was re-typed under the electropherogram with the entire 21-base reverse PCR primer underlined. The number 25088–25227 indicates the position of this segment of sequence in the SARS-CoV-2 genome.

Since the two sequences illustrated in Figure 5 are >3000 nucleotides apart within the S gene of the SARS-CoV-2 genome, their presence in one sample supported the interpretation that the sample being tested contained an intact S gene and was suitable as the material for the development of methods for S gene target specific mutation assays.

### 3.2. Limitations of the Size of Diagnostic RT-PCR Amplicon

Initially, attempts were made to design primary and nested PCR primers to amplify a 1524-base segment of the S gene, encoding the first 508 amino acids of the SARS-CoV-2 spike protein (Figure 2), including the NTD and the ACE2 RBD in a single amplicon. It has been reported that, under certain conditions, the entire >1500-base bacterial 16S rRNA gene can be amplified by PCR [24,25]. However, all attempts failed. A single >1500-bp S gene cDNA PCR amplicon could not be generated from the nasopharyngeal swab samples used for this study.

### 3.3. Target Amplicon Sequencing of the ACE2 RBD Region

According to the CDC’s definitions, all SARS-CoV-2 variants of concern and of interest contain at least one amino acid mutation in the S protein ACE2 RBD from K417 to N501 (Table 1). However, R403T has also appeared recently at the GISAID hCoV-19 S protein mutation surveillance dashboard along with other mutations for emerging variant characterization [Figure 1]. Therefore, a diagnostic base-calling electropherogram must contain a 297-base unambiguous sequence covering 99 amino acid codons (nucleotide position 22769–23065). For routine diagnostic convenience, these 297 bases must be present in one single computer-generated sequence on an electropherogram to confirm that the positive isolate is not a variant of concern or interest or to provide mutation information for variant identification. To fulfill these requirements, a pair of SS1/SS2 primary RT-PCR primers and a pair of SS3/SS4 nested PCR primers (Table 2) were selected to amplify a 460 bp primary PCR cDNA amplicon and a 437 bp nested PCR amplicon, respectively. These two pairs of primers were proven to be successful for the amplification of a 437 bp nested PCR amplicon to be used as sequencing templates from all 16 samples proven to contain a segment of N gene sequence. One of these electropherograms, showing the sequence encompassing the codons from R403 to N501, is presented in Figure 5.

**Figure 5 viruses-13-02386-f005:**
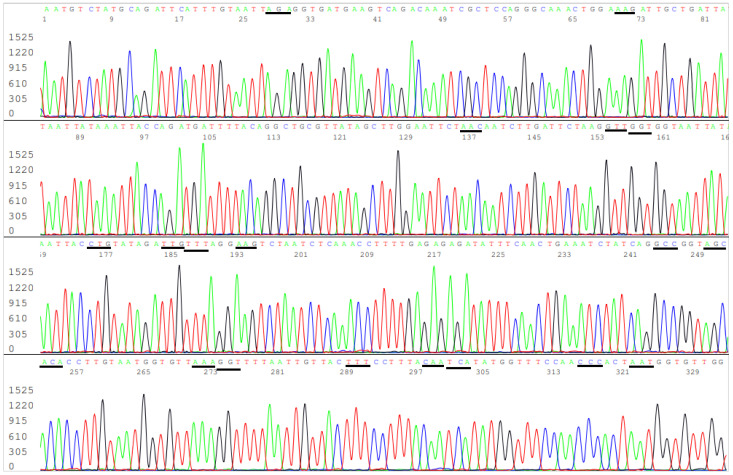
This is a figure of computer-generated electropherogram showing the codons (underlined) of 19 spike protein amino acids in the ACE2 RBD region commonly used to detect and define SARS-CoV-2 variants.

In Figure 5, the sequencing electropherogram shows 19 underlined codons of R403, K417, N439, V445, G446, L452, L455, F456, K458, A475, S477, T478, E484K (GAA > AAA mutation), G485, F490, Q493, S494, P499, and N501 in the ACE2 RBD region of the SARS-CoV-2 spike protein gene. Nonsynonymous mutations of the nucleotides in these 19 codons are routinely monitored for surveillance by GISAID (Figure 1).

### 3.4. Target Amplicon Sequencing of the NTD Region

The amino acids in the NTD region used by the CDC to define variants span from L5 to R158, a segment of 154 amino acids with a coding nucleic acid sequence of 462 bases. Attempts to generate a 569 bp nested PCR amplicon from the 16 clinical samples known to be positive for SARS-CoV-2 by partial N gene sequencing were successful in nine samples only (9/16). By necessity, the sizes of the primary RT-PCR amplicon and the nested PCR amplicon were reduced to 505 bp and 490 bp, respectively, to gain PCR sensitivity while using the SB5/SB6 pair for the primary PCR primers and the SB7/SB8 pair as the nested PCR primers to generate a 490 bp amplicon (Table 2) as the template for Sanger sequencing from all 16 samples. This 490 bp amplicon covers the codons of 17 key amino acids in a region from A67 to R158, i.e., a total of 92 codons with a 276-base sequence. Mutations in these 17 key amino acids are used by the CDC [3] to help distinguish variants of concern and interest for surveillance. Since several deletions are involved in these mutation profiles and a bi-directional Sanger may be needed for verification of some of these deletions, the size of the NTD nested PCR amplicon is longer than necessary for a one-directional reading so that the key mutation sites are not placed too close to the PCR primer sites in case a bi-directional sequencing is needed to confirm an SNP or a deletion toward the 3′ end of a nested PCR primer site. A typical computer-generated electropherogram showing the codons of the 17 amino acids in the NTD, which the CDC uses to help define variants, is presented in Figure 6.

In Figure 6, the sequencing electropherogram shows 17 underlined codons of amino acids, A67, H69, V70, G75, T76, D80, T95, D138, G142, Y144, Y145, H146, W152, E154, E156, F157, and R158 of the SARS-CoV-2 spike protein in the NTD, which may mutate in different variants of concern and of interest. These 17 amino acid mutations, along with the ACE2 RBD amino acid mutations, are used for variant categorization (Table 1).

### 3.5. Determination of Variants by Sequencing of the ACE2 RBD and NTD

Assuming the CDC’s variant classification algorithms to be valid and stringent, the profiles of the amino acid mutations listed in Table 1 can be simplified into Table 3, using combinations of the mutations in the ACE2 RBD and the NTD for accurate variant categorization.

## 4. Discussion

When RNA viruses are allowed to transmit from population to population, genetic change invariably occurs due to RNA polymerase copying errors, which may lead to single nucleotide nonsynonymous mutations and indel mutations. The wildtype Wuhan-Hu-1 SARS-CoV-2 spike protein has 1273 amino acids encoded by a 3822-base S gene. However, as of 23 August 2021, the number of S protein amino acid mutations reported worldwide already reached 2860 [26]. Even randomly mixing a small fraction of these mutations will result in an enormous number of combination profiles. Therefore, as a matter of necessity, the CDC can only select the most prevalent profiles, for example, the mutation combinations listed in the definitions of variants of concern and interest (Table 1) for analyses. However, in the United States, COVID-19 patients and their healthcare providers were not even allowed to know if the SARS-CoV-2 detected in their specimens were a Delta variant [27] because no variant test had been authorized to be used for clinical usage, and the variant surveillance tests vary greatly from laboratory to laboratory.

A university laboratory director in California was quoted as claiming that an L452R mutation is often a telling sign and that about 94% of the samples analyzed by his laboratory that show that mutation are proven to be Delta [28]. “Right now, we are assuming any new case is Delta given the high probability”, reportedly declared by an infectious disease specialist at the University of California in San Francisco [29]. It is generally believed that there is a “Lack of testing” for Delta variant and that “without adequate data, policymakers are just swinging in the dark,” as stated by a clinical professor of population and public health sciences at the University of Southern California [30]. Therefore, there is an urgent need for a science-based routine testing method for accurate detection of the key S protein amino acid mutations on all samples positive for SARS-CoV-2 so that the Delta and other variants of concern or of interest can be properly and consistently identified for further analyses.

The currently widely used whole genome/NGS technology is an emerging, not yet stable technology for general use in disease diagnosis. There is a strong opinion within the EuroGentest and the European Society of Human Genetics that, for genes that are responsible for a significant proportion of the defects, the sensitivity should not be compromised by the transition from Sanger to NGS [31]. In addition, there is a high percentage of uncertainties of base calls associated with computational errors and biases in NGS [11]. While the NGS technique is widely applied, varying error rates have been observed [32]. The first genomic sequences of SARS-CoV-2 isolates from patient specimens in China [33] and in the United States [22] were verified by Sanger sequencing to avoid base-calling errors. Since specific variant classification is based on certain key amino acid mutations in the S protein, which in turn depend on accurate determination of SNPs and indel mutations of the S gene sequence, Sanger sequencing is the method of choice if the information derived from variant testing is used to influence patient management and policy making.

Sanger sequencing needs a properly prepared template, which is usually a PCR amplicon of the target nucleic acid, for example, a segment of the S protein gene. In molecular diagnostics, the size of the PCR amplicon of the target DNA or cDNA is usually <450 bp. Attempts to amplify big-sized templates in complex samples often lead to PCR failures [34]. It is technically impossible to amplify the entire 3822-base S gene as one single amplicon to be used as a Sanger sequencing template. PCR amplification of a 405 bp fragment from the SARS-CoV genome for sequencing and comparing the sequence of the amplicon with reference sequences in the GenBank database was the established method for molecular detection of SARS-CoV during the 2003 outbreak [35,36]. The CDC’s standard diagnostic protocol for SARS-CoV recommended using three specific primers to perform heminested PCR and to sequence a 348-bp heminested PCR amplicon “to verify the authenticity of the amplified product” [37]. With accurate diagnosis, prompt isolation of patients, and early treatment, the SARS 2003 outbreak ended in June with 8098 reported cases and 774 deaths worldwide [38] without a variant of concern reported. It is of interest to note that the CDC developed a sequencing-based molecular test to facilitate ending the SARS epidemic so quickly by using just 15 positive SARS patient samples for method development and that a method for the determination of SARS-CoV-2 variants in sewage was developed by nested RT-PCR amplification of the S gene in only six samples, followed by conventional Sanger sequencing of the cDNA PCR amplicons [39]. The method presented in this article followed the CDC’s established SARS 2003 protocols [35,36,37] to sequence two ~400-base segments of the S gene of SARS-CoV-2 for accurate determination of SNP and indel mutations, which are used to determine amino acid changes to further define variants. Specific points of discussion are presented as follows.

### 4.1. Accurate Categorization of VOCs and VOIs on All Samples Positive for SARS-CoV-2

This article introduces a generic target specific mutation assay for the accurate detection of variants of concern (VOCs) and variants of interest (VOIs) by sequencing two nested RT-PCR amplicons of the SARS-CoV-2 spike protein gene, one located in the ACE2 RBD and one in the NTD region. Since the sample being tested includes the phenol-extracted digestate of virus-infected cells instead of cell-free fluid only [19,21], more viral genome copies are available for testing in this assay as compared to other commercial assays. In addition, the nested RT-PCR technology routinely amplifies the target nucleic acid for a total of 60 cycles to raise detection sensitivity. Therefore, this target specific mutation assay can determine the amino acid mutations accurately in samples with low viral loads when the whole genome/NGS surveillance technologies may fail.

Traditionally, the CDC recommends sequencing of an approximately 400-bp RT-PCR amplicon to verify the authenticity of the amplified product [35,36,37] in molecular testing for SARS-CoV. Phenol-chloroform has been shown to be a 10^6^ times more sensitive extraction method than the popular commercial QIAamp blood kit in the detection of HBV DNA in serum samples [40].

Any meaningful correlative analysis linking variants to clinical and epidemiological data must be based on precise determination of S protein amino acid mutations, which are the basis for variant categorization. Variant testing should be conducted routinely on all samples positive for SARS-CoV-2. The current surveillance programs select less than 5% of the positive samples with high viral loads for variant testing by whole genome/NGS; it is generating highly biased and potentially misleading information based on which public policy with impacts on society and economy is made. A high SARS-CoV-2 viral load in a clinical specimen is not invariably associated with disease severity [41].

### 4.2. The Current Confusion of Delta Variant Testing

When RNA viruses are allowed to pass from host to host, only those mutations that can be passed down to descendant viruses in subsequently infected individuals can be observed, documented, and reported in the literature [42]. The WHO and the U.S. CDC have selected the mutations of eight amino acids, namely K417, L452, S477, T478, E484, F490, S494, and N501, in the spike protein ACE2 RBD as the key mutations to create a limited number of VOCs and VOIs for surveillance purposes (Table 1). The WHO and CDC seem to advise that the absence of mutations in these eight amino acids rules out VOCs or VOIs, although such advice has not been clearly stated on record.

As SARS-CoV-2 continues to spread, more new amino acid mutations in the ACE2 RBD have been accumulated to the circulating strains and reported to GISAID (Figure 1). Some of these new profiles may contain a mixture of mutations, each of which is considered unique for a specific variant, such as E484K, T478K, K417T, and N501Y (Figure 1 and Table 1). It is not clear if these new profiles are considered to be VOCs or as Delta variants if there is a T478K mutation.

According to the official classification algorithms, the T478K mutation in the RBD is unique to the Delta variant; by definition, the spike protein of the Delta variant contains eight mutations, including four mutations in the NTD (T19R, G142D, 156–157del, and R158G), two in the RBD (L452R and T478K), one mutation close to the furin cleavage site (P681R) and one in the S2 region (D950N) [3,43]. The Delta variant was reported to become the most dominant SARS-CoV-2 worldwide in the summer of 2021. In the United States during the week of 22–28 August 2021, 99.1% of the SARS-CoV-2 isolates were classified as Delta variants [44].

However, the WHO’s definition for Delta variant is a profile of T19R, G142D, 157del, and 158del in the NTD plus L452R and T478K in the ACE2 RBD [12], and the Public Health England uses P681R as the key mutation to define the Delta variant [13]. It is not clear which classification algorithm is being used to define Delta variants in different parts of the world. In the United Stated, some specialists simply assumed “any new case is Delta” [29]. Since the whole genome/NGS surveillance technology tends to generate uncertainties of base calls associated with computational errors and biases [11], it is not known how many of the new cases have been erroneously classified as Delta variants as a result of computational errors and biases. Based on information available in the public domain, the sequence data for variant surveillance have not been verified by Sanger sequencing as stringently as those used to identify the initial Wuhan-Hu-1 SARS-CoV-2 strain [22,33].

### 4.3. Accumulation of Mutations in Viruses Is a Function of Passages

When RNA viruses are subjected to passages as in serial culture transfers, an accumulation of mutations will occur [45]. The same biological process takes place among humans in the current COVID-19 pandemic.

The 2003 SARS spreading ceased in June. There was no SARS-CoV variant of concern in 2003 because the epidemic ended too soon for accumulation of a significant number of mutations in the circulating viruses.

During the 2020 COVID-19 outbreak, it took 11 months for the first variant of concern, an Alpha variant of SARS-CoV-2, to develop and to be isolated from a 58-year-old human male on 24 November 2020 in England, United Kingdom [46]. An accumulation of amino acid mutations and emerging of SARS-CoV-2 variants of concern was probably the result of uncontrolled transmission of the RNA virus among populations [47].

For example, E484K is the unique mutation in combination with K417N and N501Y in the ACE2 RBD that is used to define the Beta variant, the so-called South Africa variant, first reported in December 2020 [48]. However, a search of the SARS-CoV-2 genomic sequence database in the GenBank revealed that solitary E484K mutations in the ACE2 RBD without concomitant K417N or N501Y were already reported to the GenBank from a specimen collected on 15 March 2020 in Utah, USA (Sequence ID: MW190617), from a specimen collected on 22 May 2020 in Illinois (Sequence ID: MT772530), from a specimen collected on 17 August 2020 in Utah (Sequence ID: MW420419), from a specimen collected on 28 October 2020 in Minnesota (Sequence ID: MW349062), and from two specimens collected in November 2020 in Virginia (Sequence ID: MW338781 and Sequence ID: MW411871). The same solitary E484K might have been in the SARS-CoV-2 strains circulating unknowingly in South Africa before December 2020. Some of the “Beta variant” isolates might have been locally developed in the United States.

A search of the SARS-CoV-2 genomic sequence database in the GenBank also revealed that solitary T478K mutations without concomitant mutations in the ACE2 RBD or the NTD had been reported to the GenBank on numerous occasions from different states before the summer of 2021, for example, in a specimen collected on 13 January 2021 in Utah (Sequence ID: MZ490259), in a specimen collected on 22 March 2021 in Arizona (Sequence ID: MZ914771), in a specimen collected on 24 April 2021 in Interior Alaska (Sequence ID: MZ643206), and in a specimen collected on 6 May 2021 in Anchorage, Alaska (Sequence ID: MZ927507). In addition to the solitary T478K mutation in the ACE2 RBD with a wild-type NTD sequence and a wild-type D950, the sequences mentioned above also contain a P681H mutation instead of a P681R that is used to define the Delta variant. It is not clear if these isolates are being classified as Delta variant. They are certainly not the descendent of the Delta variant originating in India. According to the currently accepted classification algorithms, P681H only occurs in the Alpha variant (Table 1).

The GenBank database contains numerous SARS-CoV-2 spike protein amino acid mutation profiles, which may be mistaken as Delta variant if a stringent variant classification algorithm is not followed. A few potential sequence profiles that can be mistaken for a Delta variant are listed as follows:L452R and T478K without concomitant mutations in the NTD (Sequence ID: MZ637393).E156del, F157del, R158G without concomitant mutations in the ACE2 RBD (Sequence ID: MZ340544).G142D, E156del, F157del, R158G without concomitant mutations in the ACE2 RBD (Sequence ID: MZ341068).T95I, G142D, E156del, F157del, R158G, E484K (Sequence ID: MZ531409).T95I, L452R (Sequence ID: MZ086521).

### 4.4. The Delta Variant Scare Is Not Supported by Facts

Currently, there is a coronavirus Delta variant scare being generated in the United States to the point that the created public anxiety may have a negative impact on the U.S. economic recovery from the pandemic [49], although even the CDC does not know exactly how many U.S. coronavirus deaths are attributable to Delta variant infections [50]. Nevertheless, according to the data published up to 5 July 2021 by Public Health England, the system recorded a total number of 170,063 cases of Delta variant infection and 259 deaths among this group of patients [51], with a mortality rate of 0.15%. In the same document, there were 225,864 cases with Alpha variant infection and 4264 deaths in the same group with a mortality rate of 1.89%. So, the Alpha variant is at least 10 times more deadly than the Delta variant.

For comparison, the Chinese data show that up to 3 March 2020, before any variants of concern emerged, there were 80,270 confirmed COVID-19 cases with 2981 deaths in China, most of which were from the epicenter of the outbreak [52]. The mortality rate of the wild-type Wuhan-Hu-1 SARS-CoV-2 infections is 2981/80,270 = 3.71%, which is about twice as high as the mortality rate of Alpha variant infections.

Therefore, the Delta variant is not more dangerous or more deadly than the wild-type Wuhan-Hu-1 strain or the Alpha variant. The high number of Delta variants being reported in the literature may have resulted from over-extrapolation bias based on sequencing of a very limited number of specially selected samples with surveillance testing methods of uncertain accuracy in unregulated laboratories. Generally, surveillance testing using sequencing technology to identify SARS-CoV-2 genetic variants can be performed in a facility that is not CLIA certified, provided that patient-specific results are not reported to (1) the individual who was tested or (2) their health care provider [53]. There are no quality control measures to identify potential flaws in coronavirus variant testing in the United States because the surveillance testing results are not for patient management, even though they are being used as the basis for the formulation of public health policies.

The high number of Delta variants being reported to the government for surveillance purposes may simply indicate that many SARS-CoV-2 strains with certain amino acid mutations described in the CDC’s definition for the Delta variant (Table 1) have acquired a genetic profile that enables them to have a higher replication rate in the host than the others, but their pathogenicity may have been reduced to the level of that of the common human coronaviruses as those of types 229E, NL63, OC43, and HKU1 [54]. This heterogeneous group of SARS-CoV-2 strains may have been detected more often because there are more virus copies in the samples being tested resulting from their higher replication rates. A higher rate of being detected does not necessarily indicate that the virus variant is more transmissible unless actual movement of the variant among close contacts has been studied by epidemiological tracing research supported by accurate variant testing. Transmissibility of a virus is primarily determined by the infectivity of the pathogen [55], not the viral load of the donor.

### 4.5. Routine Sequencing on All Positive Samples for Variant Determination

According to the CDC update for the week ending 28 August 2021, the combined proportion of cases attributed to the Delta variant is estimated to be greater than 99% in the United States. It is expected that Delta will continue to be the predominant circulating variant [56]. However, the 99% attribution to Delta is an estimate. Laboratories may use different profiles of amino acid mutations to define the Delta variant. Some reports were based on assumptions only [29].

Notably, some researchers in the field use a profile of T19R, G142D, E156G, F157del, R158del, L452R, T478K, D614G, P681R, and D950N in the spike protein to define the Delta variant by following the GISAID database [57]. According to the latter system of classification, the Delta variant lacks T95I and has E156G and R158del [57], the two mutations that are not in line with the CDC’s definition for the Delta variant (Table 1). A GenBank Sequence ID# OU534154 also lists an NTD/ACE2 RBD sequence containing G142D, E156G, F157del, R158del, L452R, and T478K with neither T19R nor T95I in the NTD. Based on the various issues discussed above, the actual number of Delta variants categorized according to the stringent CDC’s definition is unknown. All statistics based on correlations between the Delta variant and its biological characteristics are highly questionable because the SARS-CoV-2 isolates currently classified as Delta variants may actually consist of numerous genetic variants.

In order to fully realize the potential of genomic epidemiology, there is a need for routine sequencing of viral nucleic acid established in parallel with COVID-19 testing [58], on all positive samples, including those with low viral loads. Even with high viral load samples, it took several months for the CDC to accurately verify the entire ~30,000-base sequence of a SARS-Cov-2 whole genome, using both the NGS and the nested PCR/Sanger sequencing technology [22]. Such an approach, even used to sequence the entire 3822-base spike protein gene, is not practical in routine diagnostic works because the common RT-PCR amplicon size in SARS-CoV diagnostic testing is ~348 bp in size [38]. If an NGS technology is used for the diagnostic work, under certain circumstances it may need to sequence as many as 10 PCR amplicons to verify or to correct the base-calling uncertainties generated by the computational errors and biases of the NGS technology [9,11] in a gene target of 3822 bases long among numerous non-target nucleic acids in a nasopharyngeal swab sample.

This article proposes routine sequencing of a 437-bp nested PCR cDNA amplicon of the S gene ACE2 RBD (Figure 5) on all samples that are positive for a SARS-CoV-2 RNA gene. If there is no amino acid mutation in the RBD, the SARS-CoV-2 detected is not a VOC or a VOI. If the RBD sequencing shows any amino acid mutations, an additional 490-bp nested PCR cDNA amplicon of the S gene NTD is sequenced (Figure 6). Since a properly executed computer-generated sequencing electropherogram does not have ambiguous base calls, the codons of the amino acids in the ACE2 RBD and in the NTD can be easily determined without the need of bioinformatic services.

Assuming the CDC’s definitions based on amino acid mutations for variant determination to be valid and stringent, even the recently reported Mu variant (PANGO lineage B.1.621), which is characterized by a combination of R346K, E484K, N501Y, D614G, and P681H [59], can be distinguished from other VOCs and VOIs by the protocol proposed because there are no concomitant mutations in the NTD sequence in the presence of only E484K and N501Y in the ACE2 RBD sequence for the Mu variant.

By the same token, a newly reported South Africa variant with PANGO lineage C.1.2, which contains multiple substitutions (R190S, D215G, N484K, N501Y, H655Y, and T859N) and deletions (Y144del, L242-A243del) within the spike protein [60], can be distinguished from other VOCs and VOIs by the demonstration of only N484K and N501Y in the ACE2 RBD and a Y144del in the NTD without other concomitant mutations in the two amplicons targeted for Sanger sequencing.

## 5. Conclusions

The protocol presented in this article is able to sequence a 437-bp nested RT-PCR cDNA amplicon of the ACE2 RBD and a 490-bp nested RT-PCR cDNA amplicon of the N-terminal domain (NTD) of the S gene for the detection of the amino acid mutations needed for accurate determination of all variants of concern and variants of interest defined by the CDC and the WHO in samples positive for SARS-CoV-2, regardless of their viral loads. In order to fully realize the potential of genomic epidemiology, there is a need for routine diagnostic sequencing of viral nucleic acid established in parallel with COVID-19 testing on all positive samples, including those with low viral loads. Currently, there are no authorized SARS-CoV-2 variant diagnostics. In the United States, a Sanger sequencing-based variant determination assay certified under the CLIA program can be used as a routine diagnostic test for patient management and follow-ups.

## Figures and Tables

**Figure 1 viruses-13-02386-f001:**
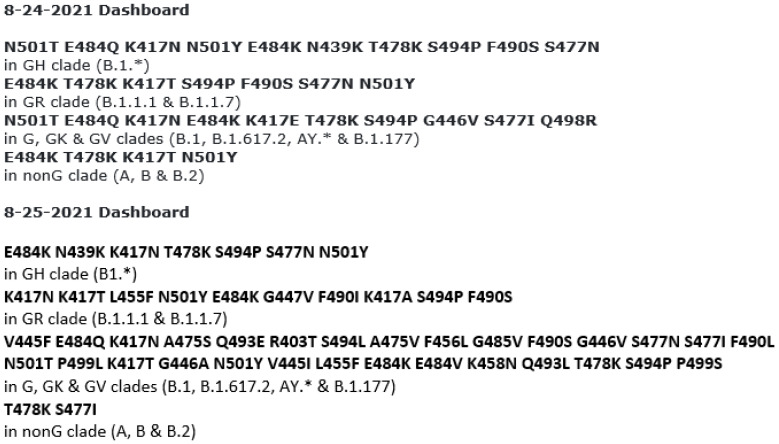
This is a figure showing GISAID hCoV-19 S protein mutation surveillance dashboard data on 24 and 25 August 2021.

**Figure 2 viruses-13-02386-f002:**
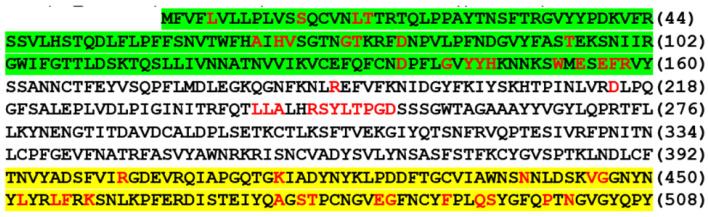
This is a figure showing the first 508 amino acids of SARS-CoV-2 S protein with highlighted NTD M1 to Y160 and ACE2 RBD T393 to Y508, retrieved from the GenBank database-Seq ID# NC_045512.2. The amino acids whose mutations (Figure 1 and Table 1) are used for variant determination are typed in red. The amino acids in the ACE2 RBD are highlighted yellow, and those in the NTD are highlighted green.

**Figure 3 viruses-13-02386-f003:**
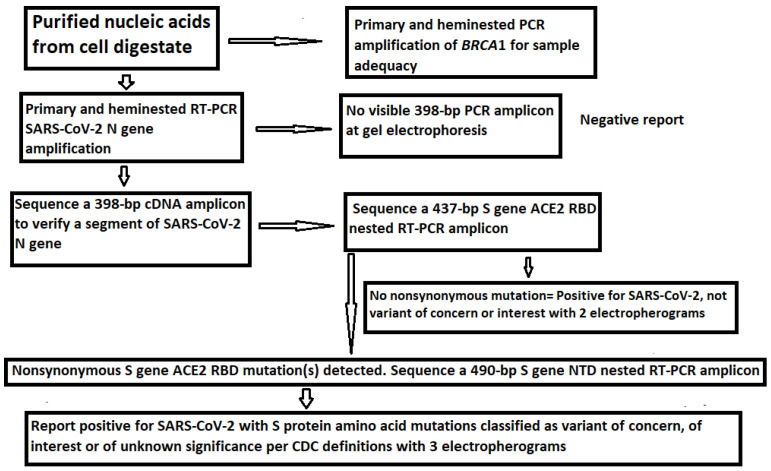
Workflow diagram for SARS-CoV-2 VOC and VOI determination by Sanger sequencing.

**Figure 4 viruses-13-02386-f004:**
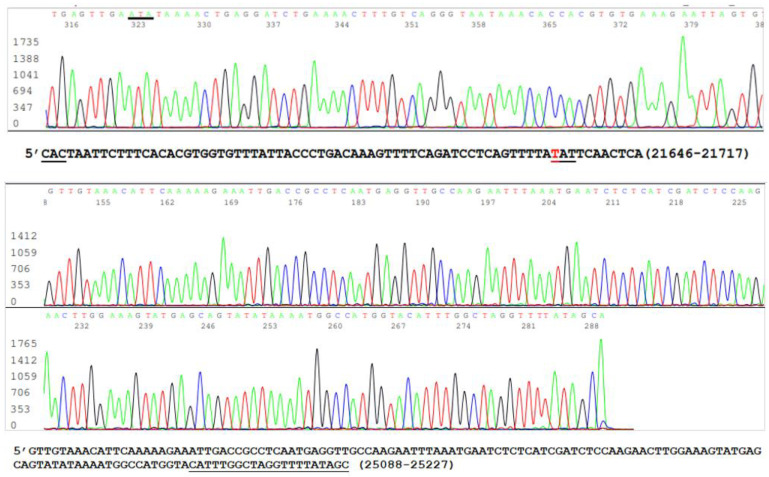
This is a figure showing two panels of sequencing electropherograms as evidence of intact S gene in a sample.

**Figure 6 viruses-13-02386-f006:**
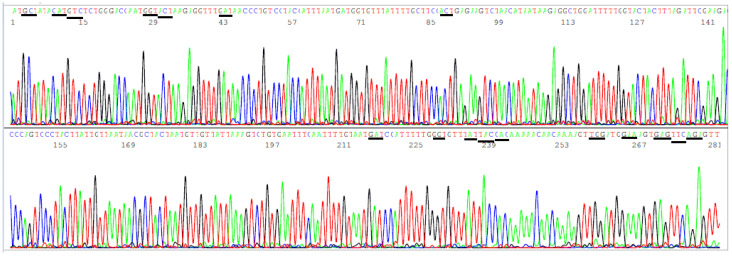
This is a figure of computer-generated electropherogram showing 17 codons (underlined) of S protein amino acids in the NTD region commonly used to help distinguish SARS-CoV-2 variants.

**Table 1 viruses-13-02386-t001:** CDC’s classification of VOCs and VOIs based on S protein amino acid (AA) mutations.

WHO Variant Labels	Constant AA Mutations	Potential Additional AA Mutations	Pango Lineage	Location of First Identification
**Alpha**	69del, 70del, 144del, N501Y, A570D, D614G, P681H, T716I, S982A, D1118H	(E484K *), (S494P *)	B.1.1.7	UK
**Beta**	D80A, D215G, 241del, 242del, 243del, K417N, E484K, N501Y, D614G, A701V		B.1.351	South Africa
**Gamma**	L18F, T20N, P26S, D138Y, R190S, K417T, E484K, N501Y, D614G, H655Y, T1027I		P.1	Japan/Brazil
**Delta**	T19R, T95I, G142D, E156-, F157-, R158G, L452R, T478K, D614G, P681R, D950N	(V70F *), (A222V *), (W258L *)	B.1.617.2	India
**Delta plus**	T19R, T95I, G142D, E156-, F157-, R158G, K417N, L452R, T478K, D614G, P681R, D950N	(V70F *), (A222V *), (W258L *)	B.1.617.2.1	India
**Epsilon**	L452R, D614G		B.1.427	California
**Epsilon**	S13I, W152C, L452R, D614G		B.1.429	California
**Eta**	A67V, 69del, 70del, 144del, E484K, D614G, Q677H, F888L		B.1.525	UK/Nigeria
**Iota**	L5F, T95I, D253G, E484K, D614G, A701V	(D80G *), (Y144- *), (F157S *), (L452R *), (S477N *), (T859N *), (D950H *), (Q957R *)	B.1.526	New York
**Kappa**	G142D, E154K, L452R, E484Q, D614G, P681R, Q1071H	(T95I*)	B.1.617.1	India
**Kappa**	T19R, G142D, L452R, E484Q, D614G, P681R, D950N		B.1.617.3	India
**Lambda**	G75V, T76I, Δ246-252, L452Q, F490S, D614G, T859N		C.37	Peru

In Table 1, the mutations in the ACE2 RBD are highlighted yellow, and those in the NTD are highlighted green. Both the CDC and the WHO have designated Alpha, Beta, Gamma, and Delta as the VOCs. The symbol (*) indicates a mutation, which may or may not occur in this variant, according to the CDC.

**Table 2 viruses-13-02386-t002:** Primary and nested PCR primers and their sequences used in this study.

S Gene Segment	Oligo-Nucleotide	Sequence	Amplicon Size BP	Location of Primer
N-terminal domain	SB5 Primary F	5′-AACCAGAACTCAATTACCCCC		21619–21639
	SB6 Primary R	5′-TTTGAAATTACCCTGTTTTCC	505	22103–22123
	SB7 Nested F	5′-TCAATTACCCCCTGCATACAC		21628–21648
	SB8 Nested R	5′-ATTACCCTGTTTTCCTTCAAG	490	22097–22117
ACE2 receptor	SS1 Primary F	5′-TGTGTTGCTGATTATTCTGTC		22643–22663
binding domain	SS2 Primary R	5′-AAAGTACTACTACTCTGTATG	460	23082–23102
	SS3 Nested F	5′-ATTCTGTCCTATATAATTCCG		22656–22676
	SS4 Nested R	5′-TACTCTGTATGGTTGGTAACC	437	23072–23092
C-terminal domain	VF1 Primary F	5′-AATCATTACTACAGACAACAC		24901–24921
	VF2 Primary R	5′-CAATCAAGCCAGCTATAAAAC	338	25218–25238
	VF3 Nested F	5′-AGACAACACATTTGTGTCTGG		24913–24933
	VF4 Nested R	5′-GCTATAAAACCTAGCCAAATG	315	25207–25227

**Table 3 viruses-13-02386-t003:** Summary of using ACE2 RBD/NTD sequencing for variant categorization. Both the CDC and the WHO have designated Alpha, Beta, Gamma, and Delta as the VOCs.

WHO Name Variant	Pango Lineage	ACE2 RBD Mutations	NTD Mutations	Location of First Identification
Alpha	B.1.1.7	N501Y	69del, 70del, 144del	UK
Beta	B.1.351	K417N, E484K, N501Y	D80A	South Africa
Gamma	P.1	K417T, E484K, N501Y	D138Y	Japan/Brazil
Delta	B.1.617.2	L452R, T478K	T95I, G142D, E156del, F157del, R158G	India
Delta plus	B.1.617.2.1	K417N, L452R, T478K	T95I, G142D, E156del, F157del, R158G	India
Epsilon	B.1.427	L452R		California, USA
Epsilon	B.1.429	L452R	W152C	California, USA
Eta	B.1.525	E484K	A67V, 69del, 70del, 144del	UK/Nigeria
Iota	B.1.526	E484K	T95I	New York, USA
Kappa	B.1.617.1	L452R, E484Q	G142D, E154K	India
Kappa	B.1.617.3	L452R, E484Q	G142D	India
Lambda	C.37	L452Q, F490S	G75V, T76I	Peru

## Data Availability

Not applicable.

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
