# Peer review of "A Routine Sanger Sequencing Target Specific Mutation Assay for SARS-CoV-2 Variants of Concern and Interest"

_viruses, 2021, doi:10.3390/v13122386_

Round 1
Reviewer 1 Report
The authors have addressed all my concerns and therefore I support publication.
However, again, I strongly suggest improving figures/tables quality as they are in low quality, while inapposite error tags under the words are still present: table 1 "pango", figure 1 "nonG" etc..
Author Response
Please see attached file "Author response to Reviewer 1 pdf"

Reviewer 2 Report
The authors present data to make a case that there is a need for a consistent and valid sequencing protocol to verify SARS CoV 2 variants, such as the Delta variant. This article introduces a protocol to sequence a 437 bp nested RT-PCR cDNA amplicon of the ACE2 RBD and a 490 bp nested RT-PCR cDNA amplicon of the N-terminal domain of the S gene for detection of aa mutations needed to determine all variants of concern and interest as defined by CDC.
To determine if this protocol was a valid approach to determine variants of concern and interest, the authors used the previously mentioned amplicons as templates for Sanger sequencing of 16 clinical specimens positive for SARS CoV 2. The authors present the case that profiles to identify the delta and other variants of this coronavirus vary from laboratory to laboratory, so there is inconsistency on how variants are defined. Basically, the data presented seems to support the concept that their protocol is a more consistent method to detect and classify variants.
The main issue with this paper is the length of the discussion. It seems to read more like a literature review and not always focused on discussing the data presented by the authors. Seems this section could be briefer and not overstate the rationale of why the authors challenged the current methods to identify delta variants of the virus. Have the authors convinced there peers that using their proposed amplicons to identify and classify variants is a or the potential standard for the industry? Also would have liked to see more than 16 samples as data points.
Overall, a reasonable well-written paper.
Author Response
Please see attached file "Author response to Reviewer 2 pdf."

Reviewer 3 Report
This article intends to introduce a generic protocol to routinely sequence a 437-bp nested RT-PCR cDNA amplicon of the ACE2 RBD and a 490-bp nested RT-PCR cDNA amplicon of the N-terminal domain of the S gene for detection of the amino acid mutations needed for accurate determination of all variants of concern and variants of interest of SARS-CoV-2. This approach including phenol/chloroform extraction of RNA, nested PCR to amplify templates, and Sanger sequencing, is accurate but time-consuming and easily contaminated. Thus, it may not be suitable for routine tests.
Several minor comments:
- Lines 72-74, Please check this sentence, [indicating a sequencing failure rate of 91% while for specimens with a Ct ≤ 27 the sequencing failure rate was only 5.3% and for those with a Ct >27 the sequencing failure rate was 75.5%].
- 3, why use BRCA1 instead of other housekeeping genes for internal control, e.g., actin, etc..
Author Response
Please see attached file "Author response to Reviewer 3 pdf"

This manuscript is a resubmission of an earlier submission. The following is a list of the peer review reports and author responses from that submission.
Round 1
Reviewer 1 Report
This manuscript describes the development of a Sanger sequencing assay to distinguish different SARS-CoV-2 variants of concern.
I agree with the author that NGS technologies need to be verified by Sanger Sequencing, especially when identification of new organisms are concerned.
Although, I do agree with the selection of the regions of interest ACE2 RDB and NTD, I have difficulties to assess the robustness of this assay, based on the testing of only 16 specimens, especially since the author recommends this assay to be used to confirm the variant in every SARS CoV-2 specimen. In addition, no specificity, sensitivity, reproducibility and precision testing has been performed.
Author Response
Dear Reviewer 1:
Please see my attached response to your kind review and helpful comments on my manuscript.
Sincerely,
The Author

Reviewer 2 Report
The manuscript entitled “A Routine Sanger Sequencing Target Specific Mutation Assay For SARS-CoV-2 Variants Of Concern And Interest” by Dr. Lee reports on a protocol to routinely sequence a 437-bp cDNA amplicon of the angiotensin-converting enzyme 2 receptor binding domain and N-terminal domain of the S protein of SARS-CoV-2 in order to accurately identify the currently known so called SARS-CoV-2 variants . Although the ms is in general too verbose this topic is off course important and interesting. This protocol would improve the identification of the different SARS-CoV-2 variants. I recommend the article for publication following an extensive reorganization/reduction of the text. I have some observations to be reviewed by authors before publishing the ms:
Thank you for letting me review this interesting work.
Major points
- I suggest including a workflow diagram/figure (better as the first one) showing the main steps of the protocol/experimental design. It would be helpful for the reader. For instance PMID: 29992095
- The first paragraphs of the methods section are too verbose. Several parts can be shortened. Similarly, the discussion section should be shortened at least by 30%. Although the topic is important, by reading the discussion, the paper seems to be more a review manuscript than a research article. Similarly, the conclusions can be shortened by reporting the main findings of the protocol, only.
Minor comments
- Line 36. This sentence is lacking in supporting references. Please include this reference PMID: 34449545
- Lines 116-117. Figure 1 should be improved for both low quality and for inapposite error tags under the words, such as “AY” and “nonG” etcs…
- Lines 130-131 and 244-246. Figures 2 and 4 should be reorganized as tables.
- Line 296. The quality of figure 5 should be improved
- Line 342 there is a period before “Figure 6”
- Line 418 SARS-CoV2. Should be SARS-CoV-2.
- Line 475 This sentence is lacking in supporting references. These notions are described in detail the following review: (https://www.mdpi.com/1999-4915/13/9/1687; DOI: 10.3390/v13091687). Please, include this reference
- Lines 608-627 the conclusions can be shortened by reporting the main findings/applications of the protocol, only.
Author Response
Dear Reviewer 2:
Please see my attached response to your kind and helpful review on my manuscript.
Sincerely,
The Author

Round 2
Reviewer 2 Report
The authors have addressed all my concerns and therefore I support publication without further changes